# A Comparative Analysis between Efficient Attention Mechanisms for Traffic Forecasting without Structural Priors

**DOI:** 10.3390/s22197457

**Published:** 2022-10-01

**Authors:** Andrei-Cristian Rad, Camelia Lemnaru, Adrian Munteanu

**Affiliations:** 1Computer Science Department, Universitatea Tehnica din Cluj-Napoca, 400027 Cluj-Napoca, Romania; 2Electronics and Informatics Department, Vrije Universiteit Brussel, 1050 Ixelles, Belgium

**Keywords:** artificial neural networks, deep learning, intelligent transportation systems

## Abstract

Dot-product attention is a powerful mechanism for capturing contextual information. Models that build on top of it have acclaimed state-of-the-art performance in various domains, ranging from sequence modelling to visual tasks. However, the main bottleneck is the construction of the attention map, which is quadratic with respect to the number of tokens in the sequence. Consequently, efficient alternatives have been developed in parallel, but it was only recently that their performances were compared and contrasted. This study performs a comparative analysis between some efficient attention mechanisms in the context of a purely attention-based spatio-temporal forecasting model used for traffic prediction. Experiments show that these methods can reduce the training times by up to 28% and the inference times by up to 31%, while the performance remains on par with the baseline.

## 1. Introduction

Traffic forecasting is concerned with predicting future values of traffic-related variables such as speed, flow, or demand, based on past observations. Applications of traffic forecasting include live route optimization, traffic simulation, or time of arrival estimation.

Tasks under the umbrella of traffic forecasting are inherently spatio-temporal. Accurate forecasting methods rely on information from multiple spatial locations and past time steps to make predictions. Consequently, state-of-the-art deep learning architectures for traffic prediction consist of building blocks that learn spatial, temporal, or joint dependencies between traffic events. The most common building blocks of these architectures are convolutional and attention layers, which are used across both temporal and spatial dimensions, and recurrent layers, which are used mostly for the temporal dimension.

Convolutional layers update the features of an element using the features of other elements within a receptive field, which is limited by the size of the learnable kernels. They are efficient due to locality and parameter sharing. Graph convolutions [1,2] are a particular class of convolutional layers that can be used to model spatial dependencies, which work well on traffic data. This is because the locations in a traffic sensor network can be represented as a graph, such that the adjacency between two nodes, corresponding to real-world locations, is proportional to their real-world distance. Models that use graph convoltuions to model spatial dependencies, such as [3,4,5], have proven to be superior than those that use regular convolutions, acclaiming state of the art on multiple datasets [6]. However these method require a connectivity prior, and they have certain shortcomings when modeling long distance dependencies.

An alternative to using graph convolutions to model spatial dependencies is to use the dot-product attention [7]. Attention layers have an unlimited receptive field and therefore can model interactions between elements regardless of their distance. This mechanism allows for entirely data-driven learning of contextual spatial dependencies in an all-to-all manner. Methods that rely entirely on attention mechanisms, such as [8,9,10] have also acclaimed state-of-the-art performance on various traffic forecasting tasks. Another advantage presented in [10] is domain adaptation, which helps to learn in scenarios with high scarcity. The main issue with the dot-product attention, when used to model spatial dependencies, is its quadratic complexity with respect to the number of sensor locations in the traffic network.

In real-world scenarios, performing accurate traffic forecasting requires the processing of very large graphs, which is computationally demanding. In addition, computing the spatial attention with quadratic complexity for very large graphs is prohibitive. The goal of this work is to identify efficient, sub-quadratic, attention mechanisms which have limited impact on the forecasting accuracy when replacing the dot-product attention. By using efficient attention modules to capture spatial dependencies, the models become easier to scale to traffic networks having thousands, or tens of thousands locations (nodes). Consequently, the main contributions of this paper are:We perform the first fair comparison between efficient attention modules in the context of a spatio-temporal forecasting model, by analyzing the performance-complexity trade-off of these modules. To our best knowledge, this is the first analysis of this type in the a spatio-temporal modeling context.We examine the results for two distinct datasets of different sizes to verify the relationship between the theoretical complexity of the attention modules and the effective training and inference times.We open-source all the data and code used in the experiments to facilitate further research in this direction.

One notes that, parallel to our work, another study [11] compares attention mechanisms in a completely different context focusing on the application of pyramid transformers for visual processing tasks. The results of this study indicate that the use of efficient attention modules only slightly impacts the performance of the baseline model, with significant gain in resource utilization.

## 2. Materials and Methods

### 2.1. Baseline Model

The baseline method considered in this work is the Attention Diffusion Network (ADN) [10]. ADN is among the state-of-the-art methods for traffic forecasting, achieving top-5 performance across multiple traffic prediction datasets such as PeMS-Bay and Metr-LA [12], and top-1 on the PeMS-07 dataset. The architecture resembles the Transformer [7], both architecturally and functionally. ADN is a stacked encoder-decoder model (Figure 1) that contains multi-head self-attention (MHA) and feedforward (FFN) layers.

Similar to the positional embeddings in the Transformer, the model projects temporal and spatial indicators into embeddings with the same hidden size as the model and adds them to the input features in the *ENC-INI* and *DEC-INI* blocks. Temporal indicators provide day-of-the-week and time-of-day encodings, while the spatial indicators attribute a unique identifier to each location without carrying any structural information.

ADN uses MHA (Equations (Equation 1) and (Equation 2)) to model both spatial and temporal dependencies. In this context, *Q*, *K* and *V* are linear projections of three input tensors. *Q* and *K* are used to generate the attention map, which is in turn used to update the features of *V*. In case of self-attention, all three are projections of the same tensor.

Normally, traffic data have three dimensions: the space, time and feature dimensions. The model is adapted in order to accommodate the additional dimension compared to regular sequential data. Unlike in other architectures such as [8,13], spatial and temporal attention are sequentially applied without an explicit merging mechanism.
(1)$${\mathrm{Attn}}_{i}\left(Q,K,V\right)=\mathrm{softmax}\left(\frac{Q\times K^{T}}{\sqrt{d_{h}}}\right)V$$
(2)$$\mathrm{Attn}\left(Q,K,V\right)=[{\mathrm{Attn}}_{0}\left(Q,K,V\right)||\ldots ||{\mathrm{Attn}}_{K}\left(Q,K,V\right)]W^{O}$$

The batched data tensor is manipulated so that the attention is applied to temporal or spatial slices. The *SPLIT* and *MERGE* blocks are responsible for splitting and merging respectively the dimensions of the data such that attention can be applied to the appropriate dimensions.

The FFN layers (not depicted) are present after each of the MHA blocks, both in the decoder and the encoder. Each FFN layer is composed of two fully-connected layers with a ReLU activation between them. Following the scheme in the transformer, the hidden internal size of the FFN layer is four times larger than the model’s hidden size. In addition to the MHA and FFN layers, the model utilizes dropout [14] and layer normalisation [15], following the regularisation scheme in the Transformer.

At training time, the model acts as an auto-encoder. The input to the encoder is a sequence of traffic states in previous time steps $P_{0},\ldots ,P_{N}$. At training time, the information about the future steps is available, as teacher forcing [16] is used. The input to the decoder is composed of the last term in the past sequence and the first N−1 terms in the future: $P_{N},F_{0},\ldots ,F_{N-1}$. The model’s output is compared with the entire future sequence $F_{0},\ldots ,F_{N}$.

At inference time, the model acts as an auto-regressor. Similarly with the training regime, the encoder uses $P_{0},\ldots ,P_{11}$ as input. The ground truth labels are no longer available, so starting with just $P_{11}$ in the decoder, the model iteratively predicts the values in the next step and adds it to the input of the decoder. After 12 steps the sequence $F_{0},\ldots F_{11}$ will be in the output of the decoder.

One of the significant benefits of the model is that it does not require spatial priors in the form of graph connectivity matrices, due to the all-to-all nature of self-attention. This allows for a completely data-driven learning, which in turn allows for better domain adaptation and more effortless transfer learning across datasets coming from different sources [10].

The downside, however, also lies in the self-attention mechanism. As it scales quadratically with respect to the number of tokens, the higher the number of spatial locations in the network, the more drastic is the effect on the training and inference time compared to other mechanisms such as convolutions or graph-based convolutions. Simultaneously, Ref. [10] introduces an alternative mechanism, which is in turn similar to the *group attention* described in [8].

### 2.2. Efficient Attention Mechanisms

As models which employ dot-product attention have seen applications ranging from natural language processing to computer vision and beyond, there has been a lot of recent interest in reducing their time and memory complexity. On the one hand, lowering the time complexity should impact the training time, but—more importantly—it could reduce the inference time, which is most often critical in real-time scenarios, while also benefiting the user experience. On the other hand, reducing complexity may impact the forecasting accuracy and the overall performance of the model. Studying this performance–complexity trade-off is the goal of this work.

Table 1 presents some alternative attention methods from the recent literature. While this study does not explicitly include a comprehensive review of sub-quadratic attention, we have selected six alternatives based on different approaches used in our experiments.

For all the methods in Table 1, *N* is the number of tokens in the sequence (number of spatial locations in our case), and *d* is the dimension of the vector representation of the tokens. For GA, M is the fixed number of partitions and K is the size of the partitions, such that N=K×M. For LA and FV, *w* and *c* are the dimensions of the lower rank projection matrices, such that c<d<<N.

#### 2.2.1. Group Attention

The group attention (GA) [10] is a form of random, local attention. The tokens are randomly partitioned into *M* groups. Each spatial location then only interacts with the other locations within the same partition. The partitions are randomly generated at each epoch.

#### 2.2.2. Reformer Attention

The reformer attention (RA) [17] uses a locality-sensitive hashing to limit the attention span. The idea behind it is that the results of the softmax in the dot-product attention are influenced, in large part, only by the most similar keys to each query.

The input sequence is transformed using a single shared weight matrix. After a hashing function is applied, the sequence is sorted by the result of the hash and chunked to a pre-defined size. The attention is then applied only to the terms within the same chunk and those with the same hash from the last bucket.

#### 2.2.3. Fast Linear Attention

The fast linear attention (FA) [18] is an approximation to the dot-product attention. Essentially, it replaces the softmax with an associative, kernel-based similarity function.

The softmax function can be generalised to any non-negative similarity function, for which the authors concretely use a kernel Φ of shape $k\left(x,y\right):R^{2\times F}⟶R_{+}$. The *i*-th row of the result is given by:(3)$$Attn{\left(Q,K,V\right)}_{i}=\frac{\sum_{j=1}^{N}sim\left(Q_{i},K_{j}\right)V_{j}}{\sum_{j=1}^{N}sim\left(Q_{i},K_{j}\right)}$$

As the softmax operation is not involved in this formulation, one can simply factor out $\Phi (Q_{i})$ since the rest of the equation is associative with respect to multiplication. $\Phi (Q_{i})$ is computed only once, and the complexity of the multiplication between the keys and the values is independent of the sequence length:(4)$$Attn{\left(Q,K,V\right)}_{i}=\frac{\sum_{j=1}^{N}\Phi {\left(Q_{i}\right)}^{T}\Phi \left(K_{j}\right)V_{j}}{\sum_{j=1}^{N}\Phi {\left(Q_{i}\right)}^{T},\Phi \left(K_{j}\right)}=\frac{\Phi {\left(Q_{i}\right)}^{T}\sum_{j=1}^{N}\Phi \left(K_{j}\right)V_{j}}{\Phi {\left(Q_{i}\right)}^{T}\sum_{j=1}^{N}\Phi \left(K_{j}\right)}$$

#### 2.2.4. Efficient Attention

The efficient attention (EA) [19] relies on the inversion of the order in which the three tensors Q, K and V are multiplied. The process starts with multiplying $K^{T}$ and *V* and applying softmax to obtain global context vectors.
(5)$$C={\mathrm{softmax}}_{col}\left(K^{T}V\right)$$

The shape of these vectors *C* is $R^{d\times d}$. As opposed to the attention map in the original model, this is independent of the number of terms in the sequence but linear with respect to the number of features, which is in most cases significantly smaller than the number of terms in the sequence and is constant with respect to the input.

To obtain the final representation, the global attention vectors are multiplied with *Q*, with an additional softmax operation.
(6)$$Attn\left(Q,K,V\right)=\frac{{\mathrm{softmax}}_{row}\left(Q\right)C}{\sqrt{d_{h}}}$$

Using only an additional softmax operation makes it possible to obtain an approximate equivalent of the quadratic attention in linear space and time.

#### 2.2.5. Linformer Attention

The linformer attention [20] proposes a low-rank approximation of the attention map, which is denoted with *P*. The authors prove that *P* can be approximated by $\tilde{P}\in R^{n\times n}$, and the error of this approximation is bounded.

Before computing and applying the attention, *K* and *V* are projected into a lower dimension using two projection matrices, $E_{i}$ and $F_{i}$, corresponding to the *i*-th head in a multi-head setting.
(7)$$Attn_{i}\left(Q,K,V\right)=\mathrm{softmax}\left(\frac{QW_{i}^{Q}{\left(E_{i}KW_{i}^{K}\right)}^{T}}{\sqrt{d_{k}}}\right)\times F_{i}VW_{i}^{V}$$

Combining the heads through the regular mechanisms is equivalent to a low-rank approximation of attention.

#### 2.2.6. FAVOR+ Attention

The fast attention via positive orthogonal random features (FV+) attention [21] proposes an approximation method similar with [18] to approximate the bottleneck softmax in the dot-product attention.

The method uses random positive-definite kernels with specific trigonometric constraints to approximate the softmax and relies on the associativity property to first perform the $K^{T}\times V$ computation.
(8)$$Attn\left(Q,K,V\right)=\frac{\Phi \left(Q\right)(\Phi {\left(K\right)}^{T}V)}{\mathrm{diag}(\Phi \left(Q\right)\left(\Phi {\left(K\right)}^{T}I_{N})\right)}$$

### 2.3. Evaluation Data

All the models were trained and evaluated on two datasets that measure traffic **speed**, with a different number of spatial locations. The datasets provide continuous measurements of average speeds over 5-minutes intervals, which are measured for a few months each. ADN constructs the dataset by splitting the data into intervals of 2 h (24 × 5-minutes intervals), with an overlap of 12 between consecutive data points. The model learns to predict the future interval (target) using the past interval (source).

The Metr-LA dataset contains average speed data from N = 207 sensors corresponding to locations on the highway network of Los Angeles, which are reported between the 1 March and the 30 June 2012. With an overlap of 12, the number of data points that can be constructed using this interval is 2855.

The PeMS-Bay dataset contains average speed data from N = 325 sensors corresponding to locations within the Bay Area, California, which are reported between the 1 January and the 31 May 2017. With an overlap of 12, the number of data points that can be constructed using this interval is 4342.

For both datasets, we follow a chronological 70/10/20 train-validation-test split. All the features are normalised via standard scaling, using the mean and standard deviation of the training set.

### 2.4. Experiment Methodology

The main goal of this study is to assess the impact of replacing dot-product attention with other efficient alternatives in a spatio-temporal deep learning architecture. All spatial attention blocks of the architecture are replaced, in turn, with one of the alternative attentions. In total, we benchmark seven models across two datasets. To measure the impact, we are interested in the performance of the models in terms of prediction performance and resource usage.

#### 2.4.1. Error Metrics

The error metrics below have been employed to measure the the performance of the models; **N** indicates the number of spatial locations, **H** indicates the number of time steps, and *y* and $\hat{y}$ represent the ground truth and the predicted values, respectively.

Mean Absolute Error (MAE)—$\frac{1}{N}\frac{1}{H}\sum_{t=1}^{H}\sum_{i=1}^{N}\left|y_{i}^{t}-{\hat{y}}_{i}^{t}\right|$Root Mean Squared Error (RMSE)—$\sqrt{\frac{1}{N}\frac{1}{H}\sum_{t=1}^{H}\sum_{i=1}^{N}{\left(y_{i}^{t}-{\hat{y}}_{i}^{t}\right)}^{2}}$Mean Absolute Percentage Error (MAPE)—$\frac{1}{N}\frac{1}{H}\sum_{t=1}^{H}\sum_{i=1}^{N}\frac{1}{y_{i}^{t}}*\left|y_{i}^{t}-{\hat{y}}_{i}^{t}\right|$

#### 2.4.2. Resource Metrics

The following complexity metrics quantify the resources used by the models:Training time (sec./epoch)—The amount of time, in seconds, required for training the model for a single epoch. This includes any epoch-level pre-processing, plus the times needed for the forward and backward passes for all the batches.Inference time (ms./epoch)—The amount of time, in milliseconds, required for inference on a single sample. This is effectively the time required to produce the output sequence for a single sample by repeatedly forward passing the data in an auto-regressive manner.Maximum GPU utilisation during training (GB)—We measure the maximum GPU utilization from a practical point of view. Knowing or approximating the maximum GPU usage, one could select an appropriate machine with a large enough GPU.

To limit the impact of the randomly selected seeds, we run each experiment 10 times and compute the mean and standard deviation of the performance metrics. To compare the alternatives with the baseline, we run an unpaired t-test and highlight those scenarios where the alternative attention mechanisms are not statistically worse than the baseline. To avoid overcrowding the result tables, we only report the mean values of the metrics.

### 2.5. Experimental Setup

For a fair comparison, all the experiments were run on an Nvidia DGX V100 workstation, using a single Tesla V100 GPU (32 GB). We closely follow the values and procedures described in [10] for the hyperparameters and training regime. Batch size: 32, maximum epochs: 100, learning rate: 0.02, learning rate decay: 15% at epochs 15, 30, 45, hidden dimension: 32, feedforward dimension: 256, dropout: 0.3, gradient clipping: 0.1.

For the model, we use our own implementation which we also open-sourced (https://github.com/radandreicristian/adn, (accessed on 6 July 2022)). For the alternative attention mechanisms, we use open-source packages and implementations (https://github.com/lucidrains/linformer, (accessed on 7 September 2022)), (https://github.com/cmsflash/efficient-attention, (accessed on 5 July 2022)), (https://github.com/idiap/fast-transformers, (accessed on 2 July 2022)), (https://github.com/lucidrains/performer-pytorch, (accessed on 10 July 2022)), (https://github.com/lucidrains/reformer-pytorch, (accessed on 18 August 2022)).

## 3. Results

The tables in this section compare the baseline with modified versions that use alternative attention mechanisms to model spatial dependencies. The naming of the models in the table follows the abbreviations in Table 1, by adding the name of the attention at the end.

### 3.1. Learning Curves

The training and validation losses for each dataset were plotted on the same chart to compare the training status visually. Figure 2 and Figure 3 show the loss curves for the Metr-LA dataset, while Figure 4 and Figure 5 show the curves for the PeMS-Bay dataset.

All the models with alternative attentions converge in the same range as the baseline. This happens both during training and validation. The validation curves are more jittery initially but then become more stable, indicating a good training regime and a robust regularisation scheme.

### 3.2. Times and Resource Utilization

Below we compare the models in terms of the resource metrics previously described. The number of parameters is consistent throughout almost all models. ADN-LSH uses parameter sharing between $W_{K}$ and $W_{Q}$ within the same block (which does not impact the performance, according to [17]), so there are fewer parameters than in the other models. ADN-LFM has a higher number of parameters due to the projection size hyperparameter. However, we consider this difference neglectable, as it accounts for no more than 3% of the total number of parameters.

For the Metr-LA dataset, Table 2 shows that the training times are usually lower than the baseline with full attention. From the training time point of view, the best alternatives (FA, LA) lower the training time by up to 23%. From the inference point of view, EA is the best alternative, outperforming the baseline with group attention by 11%. The peak GPU usage is on par with the group attention proposed by the authors.

The bottleneck of ADN-GA at train time is the chunking and indexing of the tensors to create random groups. While the complexity of this operation is linear with respect to the number of spatial locations, it is more visible when the number of locations is smaller. At inference time, this operation is only done once; consequently, the per-sample inference time is much lower than that of ADN-DA.

Looking at the training times in Table 3, ADN-FV is the best model on PeMS-Bay, taking 31% less time than the baseline. The inference time is up to 26% lower, with the best model being EA. The GPU usage is, again, on par with the group attention proposed by the authors. Larger relative differences are due to the number of spatial locations, with respect to which the dot-product attention is quadratic.

ADN-RA was, in both cases, worse than the baseline in terms of training time and peak GPU usage. This can be attributed to the additional tensor operations related to hashing and sorting. Although, in theory, its complexity is O(Nlog(N)), the sub-par results can be attributed to the practical implementation bottlenecks. As the relative difference in training and inference times is smaller for PeMS-Bay, these results would likely be alleviated by a higher number of spatial locations.

### 3.3. Prediction Performance

On Metr-LA, just a few of the alternatives achieve results that are on par with the original quadratic attention mechanism. According to Table 4 the best overall performance is achieved by the fast linear attention, which is on par with the baseline on all metrics except the short-term MAPE. Overall, the least-performing models are ADN-RA and AND-LA, which are almost always worse than the baseline.

On PeMS-Bay, according to Table 5, more alternatives are on par with the baseline. Once again, the model that is most consistently on par with the baseline is ADN-FA. Interestingly, ADN-EA performs worse in the short term, while ADN-FV performs worse in the long term.

Similarly, with Metr-LA, the short-term predictions are generally closer to the baseline. Again, this can be attributed to the autoregressive nature of the model. This phenomenon seems to manifest independently of the dataset.

## 4. Discussion

### 4.1. Theoretical Complexity vs. Training Time

Most of the alternative attention mechanisms are linear with respect to the number of tokens. However, they also depend on other parameters, such as the hidden layer dimension, and in some cases on other projection dimensions. When these other values are closer to the number of tokens N, the complexity reductions brought by the subquadratic attention mechanisms is reduced; for instance, when d≈N, the complexity becomes $\;O(N^{3})$, both for the baseline and two of the efficient attentions (**EA** and **FA**).

For cases when both *N* and *d* would be large, **LA** and **FV** attentions would be more efficient due to the constant projection size hyperparameter. In practice, it is unlikely that *d* has a magnitude larger than tens to hundreds, while the number of tokens, corresponding to real-world sensor locations, is usually in the range of hundreds and even thousands and more for large scale forecasting at country level.

From a theoretical point of view, the difference between the baseline and the efficient alternatives should be more visible on datasets with more locations. This can be seen in practice, as the relative decrease in training time is larger in case of PeMS-Bay (N = 325) −28%, than in case of Metr-LA (N = 207) −23%. Similarly, the relative decrease in inference time is larger on PeMS-Bay, 26%, than on Metr-LA, 11%.

Comparing the performance of the model on the two datasets, it is clear that the improvements in training and inference times are more significant on the larger dataset. That is, the larger the number of location, the larger the effective difference between computations with quadratic and linear complexity becomes.

### 4.2. Performance of Alternative Attention Mechanisms

One attention module (**FA**) stands out for both datasets. It performs on par with the baseline in almost all metrics, while the required training and inference times are 23%, respectively 24% less.

Another module (**FV**) is on par with the baseline on Metr-LA. On PeMS-Bay, its performance is partially on par with the baseline, for short and medium-term predictions. On long-term predictions, its performance degrades. This may indicate a limitation caused by the combination of the approximate nature of the attention and the autoregressive nature of the model. Most of the other alternative attentions have worse results due to their approximate nature. Although the difference is statistically significant, the relative difference is usually less than 5%.

### 4.3. Social Considerations

All stakeholders involved in designing and integrating large-scale AI systems, including smart transportation systems, must always take into the account computational complexity. An important consequence of training and deploying large models is their carbon footprint. Reducing complexity has a direct impact on reducing the carbon footprint of the employed models while maintaining forecasting performance.

By showing that sub-quadratic attention mechanisms can produce models that have prediction performance on par with the baseline methods for traffic forecasting, we hope to motivate further research and development to consider these as viable alternatives in the context of attention-based spatio-temporal forecasting. Although the difference in training time (and consequently the number of GPU-hours) is relatively small for the reported experiments, it can dramatically increase for larger scale networks with many more locations and/or time steps.

### 4.4. Limitations

The effect of the alternative attentions in spatio-temporal forecasting scenarios is impacted both by the model and by the data. Although it is hard to generalize from one model or dataset to another, this work demonstrates that sub-quadratic attention models can be used in traffic forecasting as they bring substantial complexity reductions with limited impact on overall forecasting performance.

## 5. Conclusions

This work conducts a comparative analysis of efficient alternatives to dot-product attention for modeling spatial dependencies in a spatio-temporal architecture for traffic forecasting. We focus on the ADN model, which originally employs dot-product attention for spatial modeling, and replace the attention mechanism with five alternatives of sub-quadratic complexity. To evaluate their performance, all the models were tested against two datasets with different number of spatial locations. In order to have a fair comparison, we use the same hardware and software settings for all experiments.

The experimental results show that some of the alternative attentions can achieve results that are on par with, or slightly better than the baseline, using significantly less computational resources. Specifically, for the architecture considered in our study, the best performing model across the two datasets is ADN-FA, which produces results on par with the baseline, while reducing the training and inference times by 25%. This makes sub-quadratic attention mechanisms extremely attractive for scenarios having thousands, or tens of thousands traffic sensors.

## Figures and Tables

**Figure 1 sensors-22-07457-f001:**
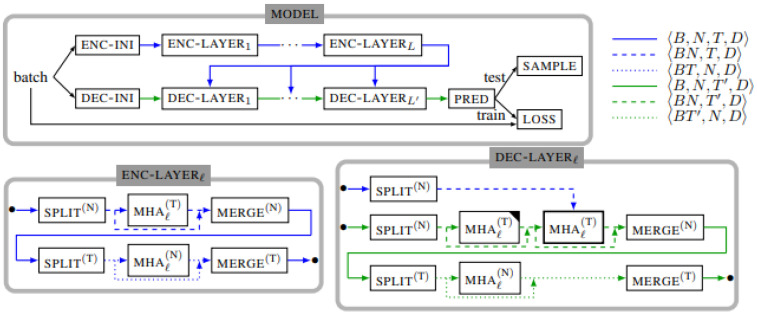
The architecture of ADN (from [10]).

**Figure 2 sensors-22-07457-f002:**
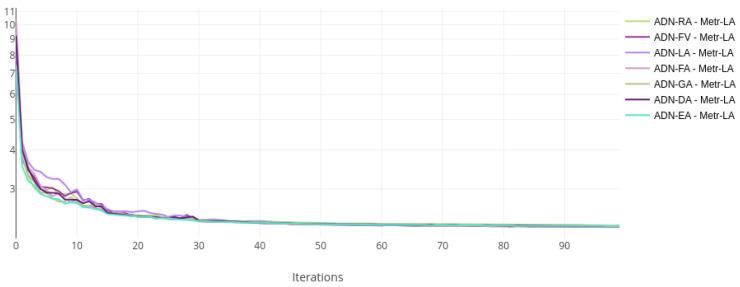
Training loss curves, log scale—ADN on Metr-LA.

**Figure 3 sensors-22-07457-f003:**
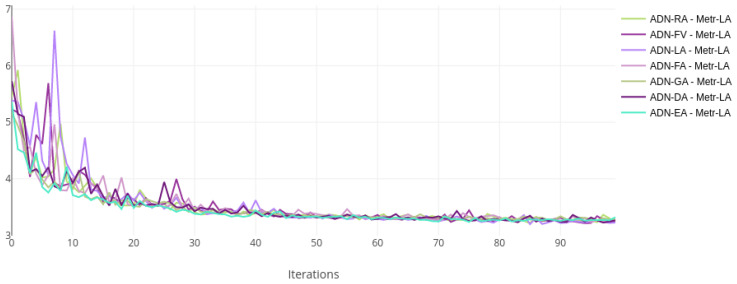
Validation loss curves—ADN on Metr-LA.

**Figure 4 sensors-22-07457-f004:**
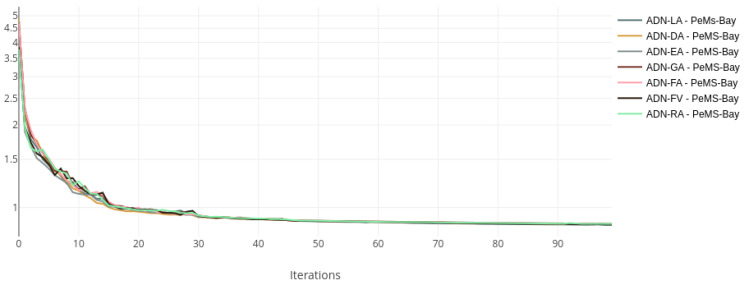
Training loss curves, log scale—ADN on PeMS-Bay.

**Figure 5 sensors-22-07457-f005:**
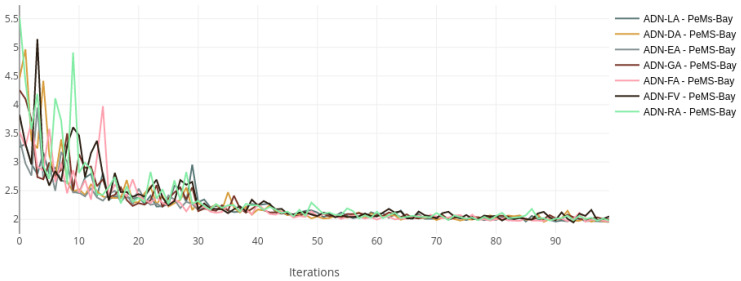
Validation loss curves—ADN on PeMS-Bay.

**Table 1 sensors-22-07457-t001:** An overview of some sub-quadratic attention strategies.

Attention Type	Complexity	Strategy
Dot-product Attention (**DA**) [7]	$O(N^{2}*d)$	All-to-all
Group Attention (**GA**) [8,10]	$O(M*K^{2}*d)$	Inter-group all-to-all
Reformer Attention (**RA**) [17]	O(Nlog(N)∗d)	Locality-sensitive hashing
Fast Linear Attention (**FA**) [18]	$O(N*d^{2})$	Kernelization, associativity
Efficient Attention (**EA**) [19]	$O(N*d^{2})$	Associativity
Linformer Attention (**LA**) [20]	O(N∗d∗w)	Low-Rank approximation
Performer Attention (**FV**) [21]	O(N∗d∗c)	Algebraic approximation

**Table 2 sensors-22-07457-t002:** Time and resource utilization of different attention types for ADN on Metr-LA.

Model	No. Parameters	Training Time (s/epoch)	Inference Time (ms/sample)	Peak GPU Usage (GB)
ADN-DA	331 K	30	7.8	9.9
ADN-GA	331 K	46	1.9	**4.7**
ADN-RA	324 K	50	8.1	11
ADN-FA	331 K	**23**	2.4	**4.5**
ADN-EA	331 K	**27**	**1.7**	5.5
ADN-LA	341 K	**23**	2.1	**4.5**
ADN-FV	330 K	25	2.2	**4.7**

**Table 3 sensors-22-07457-t003:** Time and resource utilization of different attention types for ADN on PeMS-Bay.

Model	No. Parameters	Training Time (s/epoch)	Inference Time (ms/sample)	Peak GPU Usage (GB)
ADN-DA	334 K	69	8.4	14
ADN-GA	335 K	72	5.4	**7.1**
ADN-RA	328 K	103	8.3	17
ADN-FA	335 K	**52**	**4.1**	**7.2**
ADN-EA	334 K	**50**	**4.0**	8.5
ADN-LA	356 K	**48**	**4.7**	7.9
ADN-FV	334 K	**43**	5.0	8.1

**Table 4 sensors-22-07457-t004:** Errors of models based on ADN with different attention mechanisms, on Metr-LA. Models marked with * represent our own implementation of the baseline.

	15-min			30-min			60-min		
**Model**	**MAE**	**RMSE**	**MAPE**	**MAE**	**RMSE**	**MAPE**	**MAE**	**RMSE**	**MAPE**
ADN-DA *	3.01	6.04	8.18%	3.57	7.40	10.28%	4.30	8.88	12.82%
ADN-GA *	3.06	6.14	8.37%	3.62	7.54	10.50%	4.37	**9.04**	12.98%
ADN-RA	3.04	6.10	8.25%	3.60	7.45	10.40%	**4.32**	**8.91**	12.99%
ADN-FA	**3.02**	**6.01**	8.20%	**3.56**	**7.30**	**10.22**%	**4.31**	**8.70**	**12.61%**
ADN-EA	3.06	6.11	8.33%	3.61	7.47	10.41%	4.36	**8.93**	13.08%
ADN-LA	3.05	6.14	8.20%	3.64	7.53	10.35%	4.42	**9.16**	13.03%
ADN-FV	**3.02**	**6.05**	8.20%	**3.58**	**7.42**	10.32%	4.38	**8.94**	13.75%

**Table 5 sensors-22-07457-t005:** Errors of models based on ADN with different attention mechanisms, on PeMS-Bay. Models marked with * represent our own implementation of the baseline.

	15-min			30-min			60-min		
**Model**	**MAE**	**RMSE**	**MAPE**	**MAE**	**RMSE**	**MAPE**	**MAE**	**RMSE**	**MAPE**
ADN-DA *	1.48	3.04	3.04%	1.86	4.14	4.16%	2.34	5.28	5.74%
ADN-GA *	1.51	3.07	3.10%	1.89	4.18	4.22%	2.38	5.31	5.79%
ADN-RA	**1.48**	**3.05**	**3.02%**	**1.87**	**4.15**	**4.10%**	**2.35**	**5.28**	**5.59%**
ADN-FA	**1.48**	**3.04**	**3.05%**	**1.87**	**4.12**	**4.16%**	**2.34**	**5.22**	**5.72%**
ADN-EA	1.50	3.05	3.06%	1.88	**4.14**	**4.15%**	**2.33**	**5.21**	**5.72%**
ADN-LA	1.49	3.07	3.06%	1.90	4.20	4.20%	2.41	5.42	5.84%
ADN-FV	**1.48**	**3.06**	3.05%	**1.90**	**4.18**	4.17%	2.42	5.36	**5.71%**

## Data Availability

The datasets are available at https://github.com/radandreicristian/traffic-datasets (accessed on 21 August 2022), the model-specific dataloaders are available at https://github.com/radandreicristian/pytorch_geometric/tree/feature/adn_datasets (accessed on 21 August 2022). The authors’ own implementation of the ADN model along with the efficient attentions is available at https://github.com/radandreicristian/adn (accessed on 21 August 2022). The training and evaluation pipeline is available at https://github.com/radandreicristian/traffic (accessed on 21 August 2022).

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
