# Peer review of "A Comparative Analysis between Efficient Attention Mechanisms for Traffic Forecasting without Structural Priors"

_sensors, 2022, doi:10.3390/s22197457_

Round 1
Reviewer 1 Report
The authors present in their manuscript very interesting and in my opinion also valuable results of comparison between different attention models. I have no major methodological remarks, but I have noticed a few editorial mistakes, which are listed below:
l. 182-184 – formulas for RMSE and MAPE should be swapped with each other
l. 217 - change the comma to a period at the end of the sentence
l. 255-257 – please use abbreviations consistently within the manuscript
l. 277 – typo in ‘Similarlty’
l. 285 – change to ‘(…) are respectively 23% and 24% less.’
L 288 – please use either ‘on par’ or ‘on-par’ consistently throughout the manuscript
Author Response
Dear Reviewer 1,
L.182-184 – formulas for RMSE and MAPE should be swapped with each other
Thank you for the observation. The formulats were swapped and are now correct.
L.217 - change the comma to a period at the end of the sentence
L. 277 – typo in ‘Similarlty’
L. 285 – change to ‘(…) are respectively 23% and 24% less.’
Thank you for pointing out the small typos and errors. They have been corrected.
L. 255-257 – please use abbreviations consistently within the manuscript
L. 288 – please use either ‘on par’ or ‘on-par’ consistently throughout the manuscript
We have updated the abbreviations and we are now consistently using on par throughout the manuscript.
Best regards.
Reviewer 2 Report
The aim of this paper is to realize a “comparative analysis between some efficient attention mechanisms in the context of a purely attention-based spatio-temporal forecasting model used for traffic prediction.” Finally, “experiments show that these methods can reduce the training times by up to 28% and the inference times by up to 31%, while the performance remains on par with the baseline.”
Relevance: very relevant
The paper is “very relevant” due to the following elements:
- the subject is very well defined and tangible
- the technical aspects are well justified and detailed
The paper is significant due to the following arguments:
- the authors' original contribution is very clearly explained
- the used methods are well described
From a quality point of view, the paper is technically correct. The entire manuscript is well written, and the template of the journal paper is used.
Anyway, the paper can be used by any person / expert desiring to develop an interest / to deepen the concepts in the area of intelligent transportation systems.
Further comments:
1. The main contributions and results of the paper should be added in order to underline the specificity of the study .
2. The Conclusion section can be organized much better (e.g. by addressing the research hypothesis).
3. The authors can improve the literature review with news references
(e.g. Weiwei Jiang, Jiayun Luo, Graph neural network for traffic forecasting: A survey, Expert Systems with Applications, Volume 207, 2022, 117921, ISSN 0957-4174, https://doi.org/10.1016/j.eswa.2022.117921)

Author Response
Dear Reviewer 2,
Thank you for the feedback. We have integrated your suggestions, as described below.
The main contributions and results of the paper should be added in order to underline the specificity of the study .
We have summarized our contributions in a list added at the end of section 1. This way, the specificity of the study is more clearly highlighted.
The Conclusion section can be organised much better (e.g. by addressing the research hypothesis).
We have reformulated the conclusion to include a short summary of the methodology, experiments and to highlight the findings of our comparison.
The authors can improve the literature review with news references
We have updated the literature review to include state-of-the-art traffic forecasting methods based on (graph-)convolutional networks as well as attention modules, in the second and third paragraph of the article. We have also included the survey that you mentioned, which we also read prior to our work.
Reviewer 3 Report
This paper deals with a Comparative Analysis of Efficient Attention Mechanisms for Traffic Forecasting without Structural Priors. The topic is interesting and well presented, but some parts need improvements. The introduction should include more analysis. The methodology followed to conduct this research should be better explained. E.g. in order to underline the novelty of this work, similar studies have to be cited and commented. Moreover, the bibliography is very poor. Why should this method be innovative and useful? Some application fields should be included. In line 231 the “best models” have to be cited. The conclusions section should include more results and comments.
Author Response
Dear Reviewer 3,
Thank you for the feedback. We have integrated your suggestions, as described below.
The introduction should include more analysis. Moreover, the bibliography is very poor.
We have added more background in the introduction, with a short comparison between convolutional-based and attention-based methods in traffic forecasting. Moreover, we have improved the bibliography by citing several state-of-the-art methods and a recent survey on this topic.
The methodology followed to conduct this research should be better explained. E.g. in order to underline the novelty of this work, similar studies have to be cited and commented.
We have slightly changed the first paragraph of Section 2.4, however, we consider that the methodology is very clear. In the baseline model, we replace the spatial attention block (which uses dot-product attention) with other, more efficient, attention blocks from the literature, that were not previously explored in the context of spatio-temporal forecasting. We run all the experiments on two datasets with different number of spatial locations, and look at the results through the prism of the complexity-performance trade-off. One similar study was mentioned at the end of the introduction, however we provided some more details to clarify their findings.
Why should this method be innovative and useful? Some application fields should be included.
We have included some of the generic applications of traffic forecasting in the beginning of the introduction. We have also highlighted the usefulness of the study in the end of the introduction.
In line 231 the “best models” have to be cited.
We have reformulated that paragraph and we are now explicitly mentioning the models by the names in the table.
The conclusions section should include more results and comments.
This is similar with the third point made by Reviewer #2. We have reformulated the conclusion to include a short summary of the methodology, experiments and to highlight the conclusion of our comparison.
Reviewer 4 Report
This paper considers the traffic forecasting problem with graph-based deep learning models. The goal is to identify efficient attention mechanisms for this specific problem. Overall the manuscript is well-written and the analysis is comprehensive, but there are still some problems.
1. The main contributions and key findings should be added into the end of the Introduction section in Page 2.
2. Some important surveys for spatio-temporal traffic forecasting should be added as the related work discussion.
[1] Jiang, W., & Luo, J. (2022). Graph neural network for traffic forecasting: A survey. Expert Systems with Applications, 117921.
[2] Ye, J., Zhao, J., Ye, K., & Xu, C. (2020). How to build a graph-based deep learning architecture in traffic domain: A survey. IEEE Transactions on Intelligent Transportation Systems.
[3] Lee, K., Eo, M., Jung, E., Yoon, Y., & Rhee, W. (2021). Short-term traffic prediction with deep neural networks: A survey. IEEE Access, 9, 54739-54756.
3. The mathematical symbols used in the equations and Table 1 should be formally defined, e.g., Q, K, V, N, d. It is better to summarize all the symbols in a table with their definitions, as a reference manual for readers.
4. Figures 2-5 have a low resolution. It is recommended to use vector graphics.
5. Some baseline results from the literature should be added into Table 4 and 5 for comparison, e.g., those from the above surveys.
Author Response
Dear Reviewer 4,
Thank you for the feedback. We have integrated your suggestions, as described below.
The main contributions and key findings should be added into the end of the Introduction section in Page 2.
This was also highlighted by other reviewers. We have summarised our contributions in a list added at the end of section 1. This way, the specificity of the study is more clearly highlighted.
Some important surveys for spatio-temporal traffic forecasting should be added as the related work discussion.
We have cited one of the suggested surveys, which we have read ourselves prior to our research.
The mathematical symbols used in the equations and Table 1 should be formally defined, e.g., Q, K, V, N, d. It is better to summarize all the symbols in a table with their definitions, as a reference manual for readers.
We have defined the Q K V notations in section 2.1, as they relate to the dot-product attention, and the other notations under Table 1, as they are related to the complexity of the models.
Figures 2-5 have a low resolution. It is recommended to use vector graphics.
We have replaced the figures with vector graphics (automatically generated by the framework we use for experiment tracking). We hope that the current quality is enough.
Some baseline results from the literature should be added into Table 4 and 5 for comparison, e.g., those from the above surveys.
We would like not to overcrowd the tables with results that are out of the scope of our study. We added a clarification that places the baseline model within the state-of-the-art models, in the first paragraph of section 1. You can see how exactly the baseline compares with other models in the paper that introduces the model.
Round 2
Reviewer 3 Report
The Paper has been improved by following the reviewers' suggestions.
Reviewer 4 Report
Dear authors,
Thanks for revising and re-submitting the manuscript. I am satisfied with the revisions.